# Associations between the *COMT* rs4680 Gene Polymorphism and Personality Dimensions and Anxiety in Patients with a Diagnosis of Other Stimulants Dependence

**DOI:** 10.3390/genes13101768

**Published:** 2022-09-30

**Authors:** Krzysztof Chmielowiec, Jolanta Chmielowiec, Jolanta Masiak, Aleksandra Strońska-Pluta, Małgorzata Śmiarowska, Agnieszka Boroń, Anna Grzywacz

**Affiliations:** 1Department of Hygiene and Epidemiology, Collegium Medicum, University of Zielona Góra, 28 Zyty St., 65-046 Zielona Góra, Poland; 2Second Department of Psychiatry and Psychiatric Rehabilitation, Medical University of Lublin, 1 Głuska St., 20-059 Lublin, Poland; 3Independent Laboratory of Health Promotion, Pomeranian Medical University in Szczecin, Powstańców Wielkopolskich 72 St., 70-111 Szczecin, Poland; 4Department of Pharmacokinetics and Therapeutic Drug Monitoring, Pomeranian Medical University, Powstańców Wielkopolskich 72 St., 70-111 Szczecin, Poland; 5Department of Clinical and Molecular Biochemistry, Pomeranian Medical University in Szczecin, Aleja Powstańców Wielkopolskich 72 St., 70-111 Szczecin, Poland

**Keywords:** addicted, stimulants, *COMT* gene, personality traits, genetics

## Abstract

Background: Research on the hypodopaminergic hypothesis of addictions showed that hypodopaminergic activity in males predicted the number of drugs used and is associated with drug-seeking behavior. Variant alleles may cause hypodopaminergic functioning as a result of the reduced density of dopamine receptors, decreased response to dopamine, increased dopamine clearance or metabolism in the reward system. The catechol-O-methyltransferase (*COMT*) is involved in the metabolism of dopamine. Personality traits may mediate the genetic predisposition to substance use disorders additively by various motivations associated with reward-seeking and regulating negative emotions, and also relate to self-control and environment selection. The aim of the study: The aim of this study was to investigate the association of the rs4680 polymorphism of *COMT* with personality dimensions and anxiety in patients addicted to stimulants other than cocaine (F15 according to WHO ICD-10 nomenclature) in the case of examined patients amphetamine. Methods: The study was conducted among patients addicted to stimulants other than cocaine (amphetamine). The study group included 247 patients addicted to stimulants (amphetamine) and the control group comprised 280 healthy male volunteers. The real-time PCR method was used to carry out genetic tests; personality dimensions were assessed using the standardized NEO-FFI and state and trait anxiety were assessed with STAI. All analyses were performed using STATISTICA 13. Results: The results of the 2 × 3 factorial ANOVA showed a statistically significant effect of the combined factor *COMT* rs4680 genotype on the group of patients diagnosed with other stimulants dependence/control (F_2,252_ = 3.11, *p* = 0.0465, η^2^ = 0.024). Additionally, we observed that the results of the 2 × 3 factorial ANOVA showed a statistically significant influence of the combined factor *COMT* rs4680 on the genotype in the group of patients diagnosis with other stimulants dependence/control (F_2,252_ = 6.16, *p* = 0.0024, η^2^ = 0.047). Conclusions: In our research, the polymorphism G/G *COMT* rs4680 genotype was associated with higher scores of STAI traits and STAI states in the patients dependent on amphetamine. In the control group we observed no such interactions.

## 1. Introduction

Substance use disorders are chronic, relapsing disorders. In the course of addiction neural pathways gradually undergo maladaptive changes that maintain a compulsive addictive behavior and increase the predisposition to use the drug of choice [1]. Changes in dopamine neurotransmission have been described in various types of addictions. Preclinical and clinical research supports the hypodopaminergic hypothesis of addictions and the increase in dopamine transmission as a therapeutic possibility [2]. Variant alleles may cause the hypodopaminergic functioning as a result of the reduced density of dopamine receptors, decreased response to dopamine, or increased dopamine clearance or metabolism in the reward system [3]. Research showed that, in males, the hypodopaminergic activity predicted the number of drugs used, and is associated with the drug-seeking behavior undertaken [4]. The hypodopaminergic hypothesis proposes that genetic polymorphisms may cause hypodopaminergic functioning and that it may explain the genetic predisposition to drug use in males [4]. In their research on genetic, personality, and environmental predictors of drug use in adolescents, Conner et al. [4] showed that in males, the hypodopaminergic genetic risk predicted the number of drugs used, but in females it was an environmental factor: negative life events predicted the number of used substances. This finding is also consistent with other research, which describes the results of significant differences in the predispositions to drug use between males and females [5]. This is why in our research we chose to examine only male patients with a diagnosis of other stimulants dependence according to WHO classification ICD-10, F15—in our patients, it was amphetamine dependence.

The dopaminergic system plays a significant role in drug reward. Catechol-O-methyltransferase (COMT) plays a role in the biological action and metabolism of dopamine as it catalyzes the biotransformation of catechol neurotransmitters, including dopamine. [6]. There are two forms of COMT in humans. The soluble form (S-COMT) is 50 amino acids shorter than the membrane-bound form (MB-COMT). Variant forms are generated by alternative splicing. In most tissues, S-COMT is responsible for only a tiny fraction of the overall activity of COMT [1]. The highest ratios of MB-COMT to S-COMT are found in the brain. Since MB-COMT has a higher substrate affinity for catecholamines than the soluble form, this MB-COMT may be important in regions where substrate levels are low [6,7,8,9].

The catechol-O-methyltransferase (*COMT*) gene is localized to chromosome 22q11.2. It consists of six exons, the first two of which are non-coding. *COMT* encodes two isoforms, soluble and membrane-bound COMT. Membrane-bound COMT is expressed in the neuronal tissue [10]. Regulation of catecholamines by COMT, especially dopamine levels in the brain, is crucial for modulating cognitive functions, especially in the prefrontal cortex (PFC) [11]. The *COMT* research has placed a great emphasis on the widespread non-synonymous single nucleotide polymorphism (SNP) of rs4680 (val158met), a change from G to A with a 48% Caucasian frequency of smaller alleles replacing valine (Val) with methionine (Met) at codon 158, resulting in an increase in COMT protein thermolability and, consequently, up to a 75% reduction in vitro COMT activity [12,13]. The *COMT* val158met polymorphism has been studied in a wide range of mental disorders, including schizophrenia, bipolar disorder, harm avoidance, anxiety disorders, addictions, cannabis addiction, obsessive-compulsive disorder, and attention deficit and hyperactivity disorder.

The dominant model explaining previous associations with val158 in addiction research is that for PFCs of drug users, the more effective COMT (containing valine) clears excess dopamine levels faster, possibly reducing the effect exerted by the drug. A previous research study into addiction showed links to val158met. The Substance Abuse Study by Vandenbergh et al. [14] showed that the highly active 158val was overrepresented in volunteers reporting significant multi-substance abuse. Horowitz et al. [15] found that the same allele was overrepresented in a small group of Israeli heroin addicts. Alcoholism and nicotine addiction may be associated with low activity of the met allele [16,17,18,19]. One explanation for these divergent associations with val158 in addiction is that susceptibility to some drugs (multi-substance abuse, heroin) is possibly impulsivity-mediated, while other addictions (alcohol, nicotine) are anxiety-mediated. However, the inconsistencies surrounding the COMT association and the complex phenotype and etiology of addiction show that the COMT involvement is likely to be more complicated. This may be especially true because of the well-known “U-curve” model that relates dopamine levels to the PFC function [20].

The correlation between stimulant abuse and the *COMT* Val108/158Met polymorphism has not been thoroughly investigated. As far as we know, few studies have investigated the relationship between the *COMT* gene and stimulant addiction. In the Chinese population, the *COMT* Val108/158Met polymorphism was associated with methamphetamine use, with the Val allele being a stimulant abuse risk allele [21].

Two other studies found no association between the *COMT* Val108/158Met polymorphism and methamphetamine use in participants from Japan [22] or America [23]. Tammimäki’s meta-analysis also showed no association between stimulant abuse and the *COMT* Val108/158Met polymorphism. Nevertheless, the Val allele appeared to protect against a spontaneous relapse of methamphetamine-induced psychosis [23,24]. All these studies lack an adequate statistical power.

Research on the etiology of substance use disorders associations among gene–gene and gene–nongene (e.g., psychological and environmental variables, which may have their genetic basis) interactions is being conducted [4].

Substance use disorders are now recognized as disorders in which the substance of addiction, social environment and personality interact with genetic factors influencing the neurobiology and pathophysiology of the brain [25]. Converging evidence supports moderate to high heritability, with an estimate of 30 to 70% [26]. Despite solid indications for genetic and familial influences, identifying variants of susceptibility to addiction is difficult [27].

According to the Diagnostic and Statistical Manual of Mental Disorders, 5th edition, [28] personality traits are “enduring patterns of perception, relating and thinking about the environment and oneself that are exposed in a wide range of social and personal contexts.” This approach to traits is one of the main theoretical areas in the study of personality. Studies show that personality traits are stable in people over time and in different situations and are the same in people from different cultures and languages [15,16,17]. Estimates of the heredity of the five domains proposed by Costa, P.T.; McCrae, R.R. most are commonly used to study the human personality; neuroticism (N), extraversion (E), openness to experience (O), agreeableness (A) and conscientiousness (C) are 41%, 53%, 61%, 41% and 44% heritable, respectively [29].

In a research study on personality features in amphetamine-dependent people, motor/action impulsivity, trait impulsivity, and anxiety sensitivity were specific to sibling pairs discordant for substance dependence [30,31].

The heterogeneity and complexity of substance use disorders are associated with failed molecular genetic efforts, such as genome-wide association studies (GWAS) used to locate specific genes and account for the same amount of genetic variance as twin studies. Another conceptual approach proposes an “endophenotype” approach in research on substance use disorders. Endophenotypes are measurable traits, genetically “simpler” than substance use disorders themselves. Neurocognitive functions are particularly suitable as endophenotypes and are more objective than self-reported measures. Among neurocognitive functions, neurocognitive dimensions of impulsivity have received the strongest support as a candidate endophenotype for substance use disorders, and also bipolar disorder [30], as well as traits and disorders within the externalizing and internalizing spectrum. Externalizing traits are characterized by neurocognitive deficits in impulse control; internalizing traits have been associated with reward and punishment processing abnormalities, increased loss, risk aversion, decreased delay discounting, increased attentional lapses, as well as increased negative affect. Previous research studies showed that anxious–impulsive personality traits are candidate endophenotypes for stimulant dependence [32,33,34].

In this study, we attempted to link the SNP rs4680 of the *COMT* gene with personality traits in people addicted to psychoactive substances.

## 2. Materials and Methods

### 2.1. Subjects

The research group included 247 patients addicted to stimulants (mean age = 27.58, SD = 5.75), whereas the control group included 280 healthy male volunteers matched for age (mean age = 21.99, SD = 4.30) in an interview without mental disorders (Table 1). Both groups comprised males of Caucasian origin from the same region of Poland. 

The group of patients addicted to stimulants (amphetamine and methamphetamine) included 247 male patients diagnosed with other stimulants dependence according to the International Classification of Diseases, 10th Revision (ICD-10). The patients addicted to stimulants were recruited in the Substance Use Disorder and Dependency Unit at the Residential Inpatient Treatment Center in Lubuskie, Poland, after abstaining from drugs for three months. None of the subjects received pharmacotherapy (Table 1).

The criteria for exclusion from the study in the group with patients addicted to stimulants were a medical history of psychosis (schizophrenic, affective), significant mood and/or anxiety disorders that required pharmacological treatment, and intellectual disability or genetic, severe, or uncompensated somatic (endocrinological, cardiovascular, renal, neoplastic, autoimmune, cachexia) or organic (with a manifestation of epilepsy) diseases. The healthy controls had normal intellectual skills, free of any psychoactive substances addictions (SPA) and without use of substances in a risky and harmful pattern (both present and past), as well as were free from somatic and psychic disorders.

The study was carried out at the Independent Laboratory of Health Promotion, Pomeranian Medical University in Szczecin. The protocol of the study was approved by the Bioethics Committee (KB-0012/106/16) and all participating individuals in the study were informed. Subsequently, all of them provided their written consent. All of the participants were examined by a psychiatrist and the diagnosis of other stimulants dependence was established based on the ICD10 criteria for other stimulants dependence.

The NEO Personality Inventory (NEO Five-Factor Inventory, NEO-FFI) includes 6 dimensions for each of the five traits—Extraversion (Positive Emotion, Warmth, Gregariousness, Activity, Excitement Seeking, Assertiveness), Agreeableness (Tendermindedness, Trust, Altruism, Straightforwardness, Compliance, Modesty), Openness to experience (Fantasy, Feelings, Aesthetics, Actions, Values, Ideas), Conscientiousness (Deliberation, Competence, Dutifulness, Order, Achievement striving, Self-discipline), and Neuroticism (Anxiety, Vulnerability to stress, Hostility, Self-consciousness, Impulsiveness, Depression [25]).

The results of the NEO-FFI inventories were given as sten scores. The conversion of the raw score into the sten scale was performed according to the Polish norms for adults. It was assumed that 1–2 stens were very low scores, 3–4 stens were low scores, 5–6 stens were average scores, 7–8 stens were high scores, and 9–10 stens were very high scores. 

### 2.2. Genotyping 

A standard procedure for collecting venous blood was applied to obtain genomic DNA used for genotyping in accordance with the real-time PCR method. Genotyping of rs4680 in the *COMT* gene was performed with the fluorescence resonance energy transfer in the LightCycler 480 II System (Roche Diagnostic, Basel, Switzerland) according to the standard manufacturer’s protocols.

### 2.3. Statistical Analysis

The relations between *COMT* rs4680, patients addicted to stimulants, control subjects and the NEO Five-Factor Inventory (NEO-FFI) were analyzed using a multivariate analysis of factor effects ANOVA (NEO-FFI/× genetic feature × control and patients addicted to stimulants × (genetic feature × control and Martial arts)). The homogeneity of variance was satisfied (Levene test *p* > 0.05). The distribution of the analyzed variables did not present a normal distribution. The NEO Five-Factor Inventory (Neuroticism, Extraversion, Openness, Agreeability Conscientiousness) was measured and compared using the U Mann–Whitney test. The Pearson’s correlation coefficient for age was used with the STAI ST/NEO Five-Factor scales in stimulant addicts and the control group.

The Chi-squared and Cochran-Armitage trend tests were used for an association analysis of *COMT* rs4680 and patients addicted to stimulants. The Bonferroni multiple comparisons correction was applied for these variables, and the accepted significance level was 0.0071 (0.05/7) and 0.0083 (0.05/6). All calculations were performed using STATISTICA 13 (Tibco Software Inc., Palo Alto, CA, USA) for Windows (Microsoft Corporation, Redmond, WA, USA) and CATT package in R for the Cochran–Armitage trend test.

## 3. Results

The frequency distributions were calculated with the HWE (Hardy-Weinberg equilibrium). There was no statistical deviation in the HWE between the group of patients with a diagnosis of other stimulants dependence and controls (Table 2).

The analysis of the associations of *COMT* rs4680 polymorphisms of the examined group of patients with a diagnosis of other stimulants dependence and the control group showed statistically insignificant differences in the co-dominant model in genotype frequencies for *COMT* rs4680 (G/G 24.70% vs. G/G 21.43%; G/A 47.77)% vs. G/A 45.00%, A/A 27.53% vs. A/A 33.57%, χ^2^ = 2.288; *p* = 0.3032). There were no statistically significant differences in the frequency of alleles for *COMT* rs4680 between the examined group of patients with a diagnosis of other stimulants dependence and the control group (G 48.58% vs. G 43.93%, A 51.42% vs. A 56.07%, χ^2^ = 2.288, *p* = 0.1304). Likewise, statistically insignificant differences between the control group and the group of patients with a diagnosis of other stimulants dependence were shown in the additive model (Cochran–Armitage trend test) for *COMT* rs4680 (polymorphisms Z = −1.463, *p* = 0.1433) (Table 3).

While comparing the controls and the group of patients with a diagnosis of other stimulants dependence, for the latter, we observed significantly higher scores on the STAI trait scale (M 6.78 vs. M 5.18, *p* < 0.0001), the STAI state scale (M 5.80 vs. M 4.74, *p* < 0.0001), the NEO Five-Factor Inventory scale of Neuroticism (M 6.51 vs. M 4.68, *p* < 0.0001), and the NEO Five-Factor Inventory scale of Openness (M 4.99 vs. M 4.54, *p* = 0.0078) (Table 4).

The study group compared with the control had significantly lower scores on the NEO Five-Factor Inventory scale of Extraversion (M 5.87 vs. M 6.40, *p* = 0.0045), the NEO Five-Factor Inventory scale of Agreeability (M 4.30 vs. M 5.59, *p* < 0.0001), and the NEO Five-Factor Inventory scale of Conscientiousness (M 5.58 vs. M 6.10, *p* = 0.0110) (Table 4).

In order to exclude the influence of age on personality traits, a separate correlation between age and the personality traits scale was performed for the study group and the control group. There were no statistically significant correlations (Table 5).

### 3.1. COMT rs4680 and STAI Trait Scale 

The results of the 2 × 3 factorial ANOVA showed a statistically significant effect of the combined factor *COMT* rs4680 genotype of patients with a diagnosis of other stimulants dependence/control (F_2,520_ = 5.94, *p* = 0.0028, η^2^ = 0.022) (Table 6). Power calculation: our sample had more than 88% power to detect the combined factors of those addicted to stimulants/control × *COMT* rs4680 and their interaction effect (about 2.2% of the phenotype variance). Regarding interactions, we found a significant result for the groups (addicted to stimulants vs. controls) on the STAI trait scale, and *COMT* rs4680 (F_2,520_ = 7.13, *p* = 0.0009) accounted for 2.6% of the variance (Table 6, Figure 1). The post hoc test results are included in Table 7.

### 3.2. COMT rs4680 and STAI State Scale

The results of the 2 × 3 factorial ANOVA showed a statistically significant effect of the combined factor *COMT* rs4680 genotype of patients addicted to stimulants/control (F_2,520_ = 9.25, *p* < 0.0001, η^2^ = 0.034) (Table 6). Power calculation: our sample had more than 98% power to detect the combined factor of patients addicted to stimulants/control × *COMT* rs4680 and their interaction effect (about 3.4% of the phenotype variance). Regarding interactions, we found a significant result for the groups (addicted to stimulants vs. controls) on the STAI state scale, and *COMT* rs4680 (F_2,520_ = 4.07, *p* = 0.0176) accounted for 1.5% of the variance (Table 6, Figure 2). The post hoc test results are included in Table 7.

### 3.3. COMT rs4680 and NEO FFI Neuroticism Scale 

The results of the 2 × 3 factorial ANOVA showed statistically significant interactions between the combined factor *COMT* rs4680 genotype of patients addicted to stimulants/controls and the NEO FFI neuroticism scale (F_2,520_ = 4.87, *p* = 0.0080, η^2^ = 0.018) (Table 6, Figure 3). Power calculation: our sample had more than 80% power to detect the combined addiction factor to stimulants/control × *COMT* rs4680 and their interaction effect (about 1.8% of the phenotype variance). The post hoc test results are included in Table 7.

### 3.4. COMT rs4680 and NEO FFI Extraversion Scale

The results of 2 × 3 factorial ANOVA showed statistically significant interactions between the combined factor *COMT* rs4680 genotype of patients addicted to stimulants/control and the NEO FFI extraversion scale (F_2,520_ = 4.23, *p* = 0.0149, η^2^ = 0.016) (Table 6, Figure 4). Power calculation: our sample had more than 74% power to detect the combined addiction factor to stimulants/control × *COMT* rs4680 and their interaction effect (about 1.6% of the phenotype variance). The post hoc test results are included in Table 7.

## 4. Discussion

The mesolimbic dopaminergic pathway is generally regarded as a key player in the development of addiction [34]. Undoubtedly, the mesocortical dopaminergic pathway is also involved in the onset of drug addiction, but it appears to play a more subtle and modulating role. *COMT*-knockout mice show altered dopamine clearance only in the PFC, but not in the subcortical regions [35]. Thus, the consequences of COMT disturbances are only seen in the mesocortical dopaminergic pathway. Therefore, the COMT may not have a regulatory role that is strong enough to influence the risk of developing a heavy drug dependence. Instead, it appears to be a subtle fine-tuning of the mesocortical dopamine system.

There are few studies on the potential relationship between opioid dependence and *COMT* Val108/158Met polymorphism. One study investigated the association between the *COMT* Val108/158Met polymorphism and opiate addiction in three major North American populations (Caucasian, Hispanic, and African-American) [36]. Another study on Hungarian heroin addicts found no association between the *COMT* Val108/158Met genotype and opiate use [37]. Unfortunately, both of these studies have a low statistical power. Since these studies were conducted in patients addicted to various psychoactive substances, we decided to compare them with our group, which included addicts to other stimulants. It was also important to see how these polymorphic changes ranked in our study population, by taking into account, naturally, an additional factor measured by psychometric tests.

While comparing the controls and the group of patients with the diagnosis of other stimulants dependence, for the latter, we observed significantly higher scores on the STAI trait scale, the STAI state scale, the NEO Five-Factor Inventory scale of Neuroticism, and the NEO Five-Factor Inventory scale of Openness. Previous research showed a similarity in the increase of anxiety and neuroticism in users of stimulants [38,39], which is congruent with the results obtained by Qiao and all. [40]. They used machine learning techniques to invent a model of prediction of potential drug abuse. In their research, neuroticism was the most important personality trait in detecting potential users and in estimating the usage time for stimulants. Interestingly, demographic information was less important in prediction [40]. In the previous research, high levels of openness to experience were also associated with drug use [41].

The study group compared with the controls had significantly lower scores on the NEO Five-Factor Inventory scale of Extraversion, the NEO Five-Factor Inventory scale of Agreeability, and the NEO Five-Factor Inventory scale of Conscientiousness. In the previous research, high neuroticism, low agreeableness, and low conscientiousness were consistently correlated with drug use [41]—high extraversion, in particular, was associated with cocaine/crack and stimulant use; low agreeableness was associated with cocaine/crack use [41].

The correlation between stimulant abuse and the *COMT* Val108/158Met polymorphism has not been thoroughly investigated. As far as we know, few studies have investigated the relationship between the *COMT* gene and stimulant addiction. In the Chinese population, the *COMT* Val108/158Met polymorphism was associated with methamphetamine use, with the Val allele being a stimulant abuse risk allele [21].

Obviously, we are cautious, because the cited study concerned a different population.

Two other studies found no association between the *COMT* Val108/158Met polymorphism and methamphetamine use in participants from Japan [22] or America [23]. Tammimäki’s meta-analysis also showed no association between stimulant abuse and the *COMT* Val108/158Met polymorphism. Nevertheless, the Val allele appeared to protect against a spontaneous relapse of methamphetamine-induced psychosis [22,24]. All these studies lack an adequate statistical power.

The results of the 2 × 3 factorial ANOVA showed a statistically significant effect of the combined factor *COMT* rs4680 genotype of patients addicted to stimulants/control (F_2,520_ = 5.94, *p* = 0.0028, η^2^ = 0.022). 

We found a significant result for the groups (addicted to stimulants vs. controls) on the STAI trait scale, and *COMT* rs4680 (F_2,520_ = 7.13, *p* = 0.0009) accounted for 2.6% of the variance, which was a statistically significant effect of the combined factor *COMT* rs4680 genotype of those addicted to stimulants/control (F_2,520_ = 9.25, *p* < 0.0001, η^2^ = 0.034). We found a significant result for the groups (addicted to stimulants vs. controls) on the STAI state scale, and *COMT* rs4680 (F_2,520_ = 4.07, *p* = 0.0176) accounted for 1.5% of the variance; we noticed a statistically significant interaction effect of the combined factor *COMT* rs4680 genotype of patients addicted to stimulants/control and the NEO FFI neuroticism scale (F_2,520_ = 4.87, *p* = 0.0080, η^2^ = 0.018)

The polymorphism G/G *COMT* rs4680 genotype was associated with higher severity of STAI traits and STAI states in patients dependent on other stimulants. In the control group, we observed no such interactions, which suggests that the hypodopaminergic activity in those patients may be associated with *COMT* more effectively.

The results of the 2 × 3 factorial ANOVA showed statistically significant interactions between the combined factor *COMT* rs4680 genotype of patients addicted to stimulants/control and the NEO FFI extraversion scale (F_2,520_ = 4.23, *p* = 0.0149, η^2^ = 0.016). 

In our research, we observed more significant differences between higher scores of STAI traits and STAI states and the NEO FFI neuroticism scale and lower results in Extravertism.

Our findings should be considered within the context of limitations.

First, we considered the associations between the *COMT* rs4680 gene polymorphism and personality dimensions and anxiety. This is the main limitation of our study, considering the well-known weaknesses of candidate gene studies, such as them being unpowered to detect the effects of specific variants on genetically complex traits such as personality traits or anxiety [30].

Personality dimensions and anxiety are multidimensional and multifactorial traits. They seem to be influenced by many variables—clinical and non-clinical. STAI and NEO-FFI are self-administered questionnaires with the intrinsic limitations of a subjective assessment. Other neuropsychological tests may be used to objectively assess impulsivity traits. However, they are validated instruments and are most widely used for the assessment of personality and state-trait anxiety in clinical and non-clinical populations.

We also emphasize that our study has limitations. We do not present a completely clean group of addicts because, as shown in the results, the patients had different diagnoses earlier in their medical history. We also take into account the limitations of other groups presented in the discussion, due to the statistic power. In spite of these limitations, in our study we prove beyond doubt that the direction of holistic research on addiction is the right one—taking into account biological as well as psychological and environmental factors.

## 5. Conclusions

In our research, the polymorphism G/G *COMT* rs4680 genotype was associated with higher scores of STAI traits and STAI states in patients dependent on other stimulants. However, in the control group, we observed no such interactions, which suggests that the hypodopaminergic activity in those patients may be associated with *COMT* more effectively. This finding can facilitate the development of new psychopharmacological approaches to stimulant use disorders.

## Figures and Tables

**Figure 1 genes-13-01768-f001:**
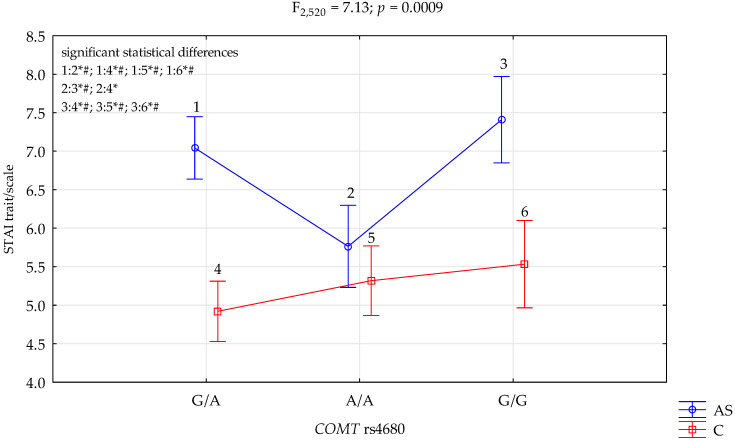
Interaction between patients addicted to stimulants (AS)/Control (C), *COMT* rs4680 and STAI trait/scale. *—significant result, # Bonferroni correction was used, and the *p*-value was reduced to 0.0083 (*p* = 0.05/6 (number of statistical tests conducted)).

**Figure 2 genes-13-01768-f002:**
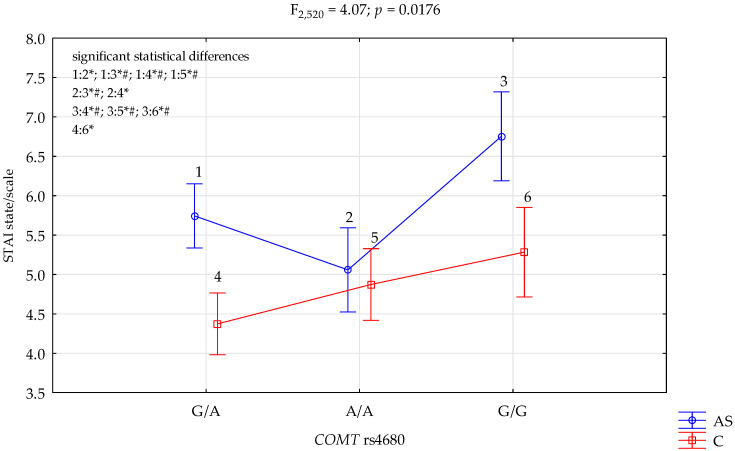
Interaction between patients addicted to stimulants (AS)/Control (C) and *COMT* rs4680 and STAI state/scale. *—significant result, # Bonferroni correction was used, and the *p*-value was reduced to 0.0083 (*p* = 0.05/6 (number of statistical tests conducted)).

**Figure 3 genes-13-01768-f003:**
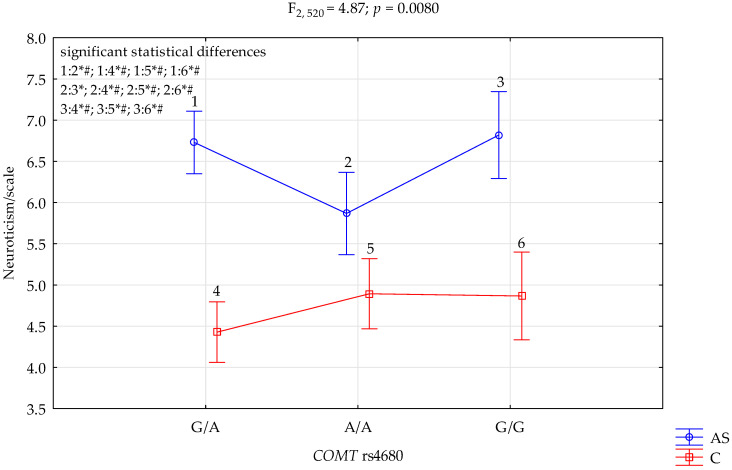
Interaction between patients addicted to stimulants (AS)/Control (C) and *COMT* rs4680 and Neuroticism scale. *—significant result, # Bonferroni correction was used, and the *p*-value was reduced to 0.0083 (*p* = 0.05/6 (number of statistical tests conducted)).

**Figure 4 genes-13-01768-f004:**
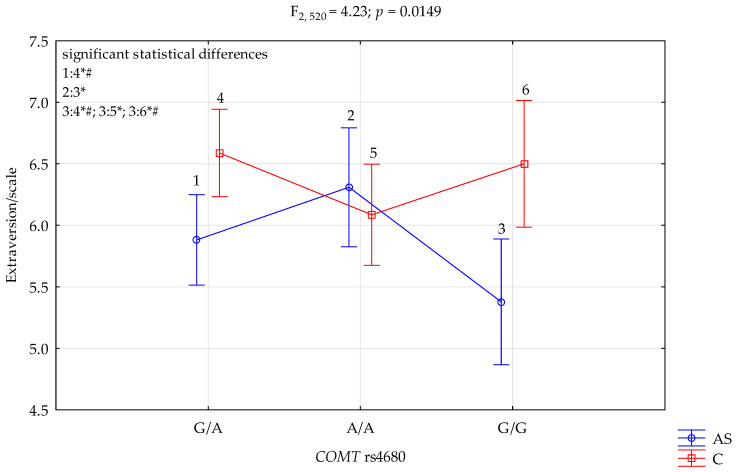
Interaction between the group of patients diagnosed with other stimulants dependence (AS)/Control (C) and *COMT* rs4680 and Extraversion scale. *—significant result, # Bonferroni correction was used, and the *p*-value was reduced to 0.0083 (*p* = 0.05/6 (number of statistical tests conducted)).

**Table 1 genes-13-01768-t001:** The comparison of the basic sociodemographic variables and coexisting psychiatric disturbances in patients addicted to stimulants and the control groups.

Characteristics		Controls *n* = 280	Addicted to Stimulants *n* = 247	*p* Value
age M(SD)		21.99 (4.30)	27.58 (5.75)	0.00000 *
education	elementary/lower secondary *n* (%)	55 (19.64%)	66 (26.72%)	0.15009
	medium *n* (%)	179 (63.93%)	146 (59.11%)	
	higher *n* (%)	46 (16.43%)	35 (14,17%)	
marital status	single	252 (90.0%)	220 (89.06%)	0.17899
	married	22 (7.86%)	15 (6.07%)	
	divorced	6 (2.14%)	12 (4.86%)	
age of the addiction onset M(SD)		NA	16.00 (3.25)	NA
duration of the addiction (yrs) M(SD)		NA	11.49 (6.06)	NA
number of the inpatient therapies M(SD)		NA	2.07 (1.74)	NA
amphetamine and methamphetamine		NA	247	
harmful drinking		NA	123 (49.80%)	NA
addiction to tobacco		NA	35 (14.17%)	NA
in the interview	depressive episode	NA	76 (30.77%)	NA
	dysthymia	NA	46 (18.62%)	NA
	hypomanic or manic episode	NA	69 (27.94%)	NA
	attempted suicide	NA	13 (5.26%)	NA
	panic disorder	NA	27 (10.93%)	NA
	agoraphobia	NA	20 (8.10%)	NA
	social phobia	NA	44 (17.81%)	NA
	obsessive compulsive disorder OCD	NA	40 (16.19%)	NA
	generalized anxiety GAD	NA	60 (24.29%)	NA
	psychotic episode(s)	NA	111 (44.94%)	NA
	anorexia	NA	2 (0.81%)	NA
	bulimia	NA	2 (0.81%)	NA

* Significant statistical differences; NA—not applicable.

**Table 2 genes-13-01768-t002:** Hardy-Weinberg equilibrium of the *COMT* rs4680 in a group of subjects with a diagnosis of other stimulants dependence and controls.

Group	*COMT* rs4680				
		Observed (Expected)	Alleles Frequency	χ^2^	*p* Value
addicted to stimulants *n* = 247	G/G	61 (58.3)	p allele freq (G) = 0.49 q allele freq (A) = 0.51	0.473	0.4916
	G/A	118 (123.4)			
	A/A	68 (65.3)			
Controls *n* = 280	G/G	60 (54.0)	p allele freq (G) = 0.44 q allele freq (A) = 0.56	2.096	0.1476
	G/A	126 (137.9)			
	A/A	94 (88.0)			

*p*—statistical significance, χ^2^—Chi^2^ test result, *n*—number of subjects.

**Table 3 genes-13-01768-t003:** Frequency of genotypes and alleles of *COMT* rs4680 in the group of patients addicted to stimulants and controls.

	Patients Addicted to Stimulants	Controls	Co-Dominant Model χ^2^ (*p* Value)	OR (95% Confidence, *p* Value)	Additive Model Cochran-Armitage Trend Test Z (*p* Value)
*COMT* rs4680					
	*n* = 247	*n* = 280	2.386 (0.3032)		−1.463 (0.1433)
G/G	61 (24.70%)	60 (21.43%)			
G/A	118 (47.77%)	126 (45.00%)		0.92 (0.60–1.42; *p* = 0.3560)	
A/A	68 (27.53%)	94 (33.57%)		0.71 (0.44–1.14; *p* = 0.0790)	
G	240 (48.58%)	246 (43.93%)	2.288 (0.1304)		
A	254 (51.42%)	314 (56.07%)			

*p*—statistical significance, χ^2^—Chi^2^ test result, *n*—number of subjects, OR—Odds Ratio.

**Table 4 genes-13-01768-t004:** STAI, NEO Five-Factor Inventory results (sten scale) between healthy controls and patients diagnosed with other stimulants dependence.

STAI/NEO Five-Factor Inventory/	Addicted to Stimulants (*n* = 247)	Control (*n* = 280)	Z	*p* Value
STAI trait/scale	6.78 ± 2.35	5.18 ± 2.21	7.302	0.0000 *#
STAI state/scale	5.80 ± 2.46	4.74 ± 2.13	5.376	0.0000 *#
Neuroticism/scale	6.51 ± 2.19	4.68 ± 2.04	9.030	0.0000 *#
Extraversion/scale	5.87 ± 2.12	6.40 ± 1.98	−2.838	0.0045 *#
Openness/scale	4.99 ± 1.99	4.54 ± 1.61	2.659	0.0078 *
Agreeability/scale	4.30 ± 1.89	5.59 ± 2.08	−7.009	0.0000 *#
Conscientiousness/scale	5.58 ± 2.27	6.10 ± 2.17	−2.544	0.0110 *

*p*—statistical significance U Mana’s test, *n*—number of subjects, M ± SD—Mean ± Standard Deviation. * Significant statistical differences. # Bonferroni correction was used, and the *p*-value was reduced to 0.0071 (*p* = 0.05/7 (number of statistical tests con-ducted)).

**Table 5 genes-13-01768-t005:** Pearson’s correlation coefficient for age with STAI ST/NEO Five-Factor scales in stimulant addicts and controls.

	STAI Trait/Scale	STAI State/Scale	Neuroticism/Scale	Extraversion/Scale	Openness/Scale	Agreeability/Scale	Conscientiousness/Scale
age addicted to stimulants R (*p* Value)	−0.0882 (0.170)	−0.0065 (0.919)	−0.0012 (0.985)	−0.0642 (0.319)	0.1142 (0.076)	−0.1019 (0.113)	−0.0829 (0.198)
age Control R (*p* Value)	−0.0560 (0.347)	−0.0774 (0.194)	−0.0466 (0.434)	0.0202 (0.735)	0.0616 (0.301)	−0.1154 (0.052)	−0.0121 (0.839)

R—Pearson’s correlation coefficient, Bonferroni correction was used, and the *p*-value was reduced to 0.0071 (*p* = 0.05/7 (number of statistical tests con-ducted)).

**Table 6 genes-13-01768-t006:** The results of 2 × 3 factorial ANOVA for patients diagnosed with other stimulants dependence and controls for the STAI and NEO Five-Factor Inventory scale, and *COMT*.

NEO Five-Factor Inventory	Group	*COMT* rs4680			ANOVA			
		G/A *n* = 243 M ± SD	A/A *n* = 162 M ± SD	G/G *n* = 121 M ± SD	Factor	F (*p* Value)	η^2^	Power (Alfa = 0.05)
STAI trait/scale	Addicted to stimulants; *n* = 247	7.04 ± 2.32	5.76 ± 2.31	7.41 ± 2.12	intercept AS/control *COMT* AS/control × *COMT*	F_1,520_ = 3464.21 (*p* < 0.0001) *# F_1,520_ = 52.82 (*p <* 0.0001) *# F_2,520_ = 5.94 (*p* = 0.0028) *# F_2,520_ = 7.13 (*p* = 0.0009) *#	0.869 0.092 0.022 0.026	1.000 1.000 0.878 0.931
	Control; *n* = 280	4.92 ± 2.23	5.32 ± 2.15	5.53 ± 2.21				
STAI state/scale	Addicted to stimulants; *n* = 247	5.74 ± 2.38	5.06 ± 2.07	6.75 ± 2.01	intercept AS/control *COMT* AS/control × *COMT*	F_1,520_ = 2730.9 (*p* < 0.0001) *# F_1,520_ = 24.32 (*p* < 0.0001) *# F_2,520_ = 9.25 (*p* = 0.0001) *# F_2,520_ = 4.07 (*p* = 0.0176) *	0.840 0.045 0.034 0.015	1.000 0.998 0.977 0.723
	Control; *n* = 280	4.37 ± 2.07	4.87 ± 2.14	5.28 ± 2.11				
Neuroticism/scale	Addicted to stimulants; *n* = 247	6.73 ± 2.15	5.87 ± 2.37	6.82 ± 1.90	intercept AS/control *COMT* AS/control × *COMT*	F_1,520_ = 3435.83 (*p* < 0.0001) *# F_1,520_ = 83.13 (*p <* 0.0001) *# F_2,520_ = 1.67 (*p* = 0.1897) F_2,520_ = 4.87 (*p* = 0.0080) *	0.868 0.138 0.006 0.018	1.000 1.000 0.352 0.802
	Control; *n* = 280	4.43 ± 1.99	4.89 ± 1.99	4.87 ± 2.22				
Extraversion/scale	Addicted to stimulants; *n* = 247	5.88 ± 2.04	6.31 ± 2.10	5.38 ± 2.20	intercept AS/control *COMT* AS/control × *COMT*	F_1,520_ = 4375.96 (*p* < 0.0001) *# F_1,520_ = 8.35 (*p* = 0.0040) *# F_2,520_ = 0.91 (*p* = 0.4042) F_2,520_ = 4.23 (*p* = 0.0149) *	0.894 0.016 0.003 0.016	1.000 0.822 0.207 0.741
	Control; *n* = 280	6.59 ± 2.16	6.09 ± 1.77	6.50 ± 1.85				
Openness/scale	Addicted to stimulants; *n* = 247	4.93 ± 1.97	5.41 ± 2.17	4.66 ± 1.75	intercept AS/control *COMT* AS/control × *COMT*	F_1,520_ = 3378.3 (*p* < 0.0001) *# F_1,520_ = 9.14 (*p* = 0.0026) *# F_2,520_ = 2.77 (*p* = 0.0635) F_2,520_ = 0.98 (*p* = 0.3753)	0.866 0.017 0.010 0.004	1.000 0.855 0.545 0.221
	Control; *n* = 280	4.59 ± 1.69	4.59 ± 1.57	4.33 ± 1.49				
Agreeability/scale	Addicted to stimulants; *n* = 247	4.24 ± 1.87	4.49 ± 1.99	4.23 ± 1.84	intercept AS/control *COMT* AS/control × *COMT*	F_1,520_ = 2943.4 (*p* < 0.0001) *# F_1,520_ = 48.332 (*p* < 0.0001) *# F_2,520_ = 0.26 (*p* = 0.7693) F_2,520_ = 0.18 (*p* = 0.8299)	0.850 0.085 0.001 0.001	1.000 1.000 0.091 0.079
	Control; *n* = 280	5.60 ± 2.12	5.61 ± 2.10	5.55 ± 2.01				
Conscientiousness/scale	Addicted to stimulants; *n* = 247	5.49 ± 2.32	5.62 ± 2.32	5.72 ± 2.16	intercept AS/control *COMT* AS/control × *COMT*	F_1,520_ = 3323.6 (*p* < 0.0001) *# F_1,520_ = 5.298 (*p* = 0.0218) F_2,520_ = 0.01 (*p* = 0.9998) F_2,520_ = 0.47 (*p* = 0.6266)	0.864 0.010 0.000001 0.002	1.000 0.631 0.050 0.127
	Control; *n* = 280	6.20 ± 2.13	6.06 ± 2.09	5.97 ± 2.36				

*—significant result; M ± SD—mean ± standard deviation. # Bonferroni correction was used, and the *p*-value was reduced to 0.0071 (*p* = 0.05/7 (number of statistical tests conducted)).

**Table 7 genes-13-01768-t007:** Post hoc LSD (least significant difference) analysis of interactions between patients addicted to stimulants (AS)/-Control (C) and *COMT* rs4680, the STAI trait scale, STAI state scale, NEO FFI neuroticism scale and NEO FFI extraversion scale.

***COMT* rs4680 and STAI trait scale**						
	{1} M = 7.04	{2} M = 5.76	{3} M = 7.41	{4} M = 4.92	{5} M = 5.32	{6} M = 5.53
(AS) *COMT* G/A {1}		0.0002 *#	0.2987	<0.0001 *#	<0.0001 *#	<0.0001 *#
(AS) *COMT* A/A {2}			<0.0001 *#	0.0124 *	0.2109	0.5591
(AS) *COMT* G/G {3}				<0.0001 *#	<0.0001 *#	<0.0001 *#
(C) *COMT* G/A {4}					0.1913	0.0810
(C) *COMT* A/A {5}						0.5621
(C) *COMT* G/G {6}						
***COMT* rs4680 and STAI state scale**						
	{1} M = 5.74	{2} M = 5.06	{3} M = 6.75	{4} M = 4.37	{5} M = 4.87	{6} M = 5.28
(AS) *COMT* G/A {1}		0.0459 *	0.0045 *#	<0.0001 *#	0.0052 *#	0.1969
(AS) *COMT* A/A {2}			<0.0001 *#	0.0427 *	0.6018	0.5723
(AS) *COMT* G/G {3}				<0.0001 *#	<0.0001 *#	0.0003 *#
(C) *COMT* G/A {4}					0.1031	0.0100 *
(C) *COMT* A/A {5}						0.2681
(C) *COMT* G/G {6}						
***COMT* rs4680 and NEO FFI neuroticism scale**						
	{1} M = 6.73	{2} M = 5.87	{3} M = 6.82	{4} M = 4.43	{5} M = 4.89	{6} M = 4.87
(AS) *COMT* G/A {1}		0.0072 *#	0.7835	<0.0001 *#	<0.0001 *#	<0.0001 *#
(AS) *COMT* A/A {2}			0.0103 *	<0.0001 *#	0.0037 *#	0.0072 *#
(AS) *COMT* G/G {3}				<0.0001 *#	<0.0001 *#	<0.0001 *#
(C) *COMT* G/A {4}					0.1041	0.1833
(C) *COMT* A/A {5}						0.9380
(C) *COMT* G/G {6}						
***COMT* rs4680 and NEO FFI extraversion scale**						
	{1} M = 5.88	{2} M = 6.31	{3} M = 5.38	{4} M = 6.59	{5} M = 6.08	{6} M = 6.50
(AS) *COMT* G/A {1}		0.1673	0.1159	0.0069 *#	0.4683	0.0552
(AS) *COMT* A/A {2}			0.0095 *	0.3625	0.4892	0.5953
(AS) *COMT* G/G {3}				0.0001 *#	0.0344 *	0.0025 *#
(C) *COMT* G/A {4}					0.0701	0.7841
(C) *COMT* A/A {5}						0.2168
(C) *COMT* G/G {6}						

*—significant statistical differences, M—mean, for these variables, # Bonferroni correction was used, and the *p*-value was reduced to 0.0083 (*p* = 0.05/6 (number of statistical tests conducted)).

## Data Availability

Not applicable.

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
