# Peer review of "Associations between the COMT rs4680 Gene Polymorphism and Personality Dimensions and Anxiety in Patients with a Diagnosis of Other Stimulants Dependence"

_genes, 2022, doi:10.3390/genes13101768_

Round 1

Reviewer 1 Report

1. The manuscript needs extensive revision for language and grammar.

2. The authors should explain the meaning of the tern “Other stimulants dependence” and describe the statistical data of stimulants used by the patients.

Author Response

Dear Reviewer,

Thank you very much for your review and valuable comments. We analyzed all the comments and replied to each, indicating where and how the corrections in the Manuscript were made, indicating the line and page.

Below are the point-by-point answers.

With respect

Authors

Comments and Suggestions for Authors

  1. The manuscript needs extensive revision for language and grammar.
  2. The authors should explain the meaning of the tern “Other stimulants dependence” and describe the statistical data of stimulants used by the patients.

- Thank you very much for your comments. We have revised the manuscript language, with the help of a native speaker. The term “Other stimulants dependence” is based on the ICD 10 WHO criteria code F15. “Other stimulants dependence” which includes mental and behavioral disorders caused by the use of stimulants other than cocaine (includes amphetamine and caffeine) as well as described the statistical data of stimulants used by the patients.

We explained what we meant by “other stimulants”. In the part material and methods, it is now completed “The group of patients addicted to stimulants (amphetamine and methamphetamine)” (…) and in Table 1.

Reviewer 2 Report

The manuscript is overall interesting and assesses an important topic, namely the impact of genetic variations on personality traits and substance abuse. The topic is interesting and worth investigating; however, it is unlikely that a single polymorphism might have a huge impact on complex aspects such as personality traits.

I have some concerns and some issue that should be resolved in order to improve the quality of the manuscript.

1- The information regarding the sample is very poor and should be improved. For example, there is no mention of what type of stimulants they used and comorbidities (e.g., personality disorders already diagnosed, other psychiatric disorders) – could this have an impact on the analyses? I suggest a table presenting the sample, also with some sociodemographic variables (status, smoking habits, job, income, etc), if they are available. Also, significant differences in these sociodemographic aspects should be reported (starting from the differences in age: it is not sufficient to report that they are “matched”)

In this sense, no inclusion vs exclusion criteria are reported. I suggest adding them

2- Similarly, the description of the analyses should be improved: for example, some confounders/covariates were controlled? Could this have an impact on the analyses?

3- I suggest expanding the literature and explaining some aspects a bit better. For example, the authors did not report very recent literature on the topic (e.g, https://www.mdpi.com/2073-4425/13/3/482 that also explore the role of gender on impulsivity, something that is very close related to substance abuse!); similarly, other references are not adequately explained: e.g., Conner et al., 2010, that is reported as one of the fundamental references to the present research, it is actually on a sample of adolescents; the difference in age range should be discussed or at least reported, in order to make the manuscript clearer.

4- A limitation section is totally missing. However, the manuscript has several limitations that should be assessed and discussed, starting from the sample and the type of drug used – it is difficult to generalize to other substance use disorders

5- Similarly, I suggest expanding the section at the end of the Introduction and creating a section devoted to aims, hypothesis, and expected results that is missing.

6- The authors reported a few times that other studies “suffer from a lack of adequate statistical power”, however they did not explain why such a statement. I suggest discussing such aspects when they refer to it.

Some minor remarks:

- Acronyms should always be reported in full length the first time they appear. HWE (Hardy-Weinberg Equilibrium) is not reported in explicit terms the first time it appears.

- I strongly suggest to fully revise the manuscript language, with the help of a native speaker, since many typos are present (e.g., in the abstract there is written “cotrol” instead on “control”; similarly, “suggest” instead of “suggest” in the Discussion etc), and some sentences are really difficult to follow. I’m not a native speaker myself and therefore I understand the difficulty of writing in a different language!

Author Response

Dear Reviewer,

Thank you very much for your review and valuable comments. We analyzed all the comments and replied to each, indicating where and how the corrections in the Manuscript were made, indicating the line and page.

Below are the point-by-point answers.

With respect

Authors

Comments and Suggestions for Authors

The manuscript is overall interesting and assesses an important topic, namely the impact of genetic variations on personality traits and substance abuse. The topic is interesting and worth investigating; however, it is unlikely that a single polymorphism might have a huge impact on complex aspects such as personality traits.

I have some concerns and some issue that should be resolved in order to improve the quality of the manuscript.

1- The information regarding the sample is very poor and should be improved. For example, there is no mention of what type of stimulants they used and comorbidities (e.g., personality disorders already diagnosed, other psychiatric disorders) – could this have an impact on the analyses? I suggest a table presenting the sample, also with some sociodemographic variables (status, smoking habits, job, income, etc), if they are available. Also, significant differences in these sociodemographic aspects should be reported (starting from the differences in age: it is not sufficient to report that they are “matched”)

In this sense, no inclusion vs exclusion criteria are reported. I suggest adding them

2- Similarly, the description of the analyses should be improved: for example, some confounders/covariates were controlled? Could this have an impact on the analyses?

3- I suggest expanding the literature and explaining some aspects a bit better. For example, the authors did not report very recent literature on the topic (e.g, https://www.mdpi.com/2073-4425/13/3/482 that also explore the role of gender on impulsivity, something that is very close related to substance abuse!); similarly, other references are not adequately explained: e.g., Conner et al., 2010, that is reported as one of the fundamental references to the present research, it is actually on a sample of adolescents; the difference in age range should be discussed or at least reported, in order to make the manuscript clearer.

4- A limitation section is totally missing. However, the manuscript has several limitations that should be assessed and discussed, starting from the sample and the type of drug used – it is difficult to generalize to other substance use disorders

5- Similarly, I suggest expanding the section at the end of the Introduction and creating a section devoted to aims, hypothesis, and expected results that is missing.

6- The authors reported a few times that other studies “suffer from a lack of adequate statistical power”, however they did not explain why such a statement. I suggest discussing such aspects when they refer to it.

Some minor remarks:

- Acronyms should always be reported in full length the first time they appear. HWE (Hardy-Weinberg Equilibrium) is not reported in explicit terms the first time it appears.

- I strongly suggest to fully revise the manuscript language, with the help of a native speaker, since many typos are present (e.g., in the abstract there is written “cotrol” instead on “control”; similarly, “suggest” instead of “suggest” in the Discussion etc), and some sentences are really difficult to follow. I’m not a native speaker myself and therefore I understand the difficulty of writing in a different language!

-We have improved the information regarding the sample and inclusion and exclusion criteria are reported. These criteria are detailed in the part “material and methods”. We hope that it is complete now.

-We have improved the description of the analyses and we have expanded the literature and a limitation section.

-The manuscript was supervised with the extensive revision for language and grammar.

Round 2

Reviewer 1 Report

After a search on Google scholar, I surprisingly found that the ethics number “KB-0012/106/16” appeared in a dozen of articles published by the same authors (Genes, 2021, 12(12): 1977.// Genes, 2021, 12(11): 1834.// Brain Sciences, 2020, 10(5): 262.// International Journal of Environmental Research and Public Health, 2018, 15(10): 2076.// International Journal of Environmental Research and Public Health, 2019, 16(15): 2687.// International Journal of Environmental Research and Public Health, 2022, 19(16): 9955.// Brain sciences, 2020, 10(6): 400.// Journal of Clinical Medicine, 2020, 9(11): 3593.// International journal of environmental research and public health, 2020, 17(1): 365.// International Journal of Environmental Research and Public Health, 2022, 19(14): 8602.// Annals of Agricultural and Environmental Medicine, 2020, 27(2).// International Journal of Environmental Research and Public Health, 2022, 19(17): 10478.). These prior articles are almost all related to “relationship between gene polymorphisms and personality traits in addicted subjects”. Patients addicted to “Polysubstance, New Psychoactive Substance, Cannabis, Heroin, Nicotine, or E-Cigarette” were reported in these prior articles. I am puzzled and wonder about the detailed design of this clinical trial. It is better for the authors to provide the approved documents of the clinical trial “KB-0012/106/16”.

Author Response

After a search on Google scholar, I surprisingly found that the ethics number “KB-0012/106/16” appeared in a dozen of articles published by the same authors (Genes, 2021, 12(12): 1977.// Genes, 2021, 12(11): 1834.// Brain Sciences, 2020, 10(5): 262.// International Journal of Environmental Research and Public Health, 2018, 15(10): 2076.// International Journal of Environmental Research and Public Health, 2019, 16(15): 2687.// International Journal of Environmental Research and Public Health, 2022, 19(16): 9955.// Brain sciences, 2020, 10(6): 400.// Journal of Clinical Medicine, 2020, 9(11): 3593.// International journal of environmental research and public health, 2020, 17(1): 365.// International Journal of Environmental Research and Public Health, 2022, 19(14): 8602.// Annals of Agricultural and Environmental Medicine, 2020, 27(2).// International Journal of Environmental Research and Public Health, 2022, 19(17): 10478.). These prior articles are almost all related to “relationship between gene polymorphisms and personality traits in addicted subjects”. Patients addicted to “Polysubstance, New Psychoactive Substance, Cannabis, Heroin, Nicotine, or E-Cigarette” were reported in these prior articles. I am puzzled and wonder about the detailed design of this clinical trial. It is better for the authors to provide the approved documents of the clinical trial “KB-0012/106/16”.

Thank you that the reviewer points this out. For us, the issues in relation to the bioethics committee and consents given by the patients are very important. Indeed, there are several articles with the same bioethics committee number–KB-0012/106/16. This is because all of these publications were funded by the research project that was selected in a national competition and received funding under contract No. UMO2015/19/B/NZ7/03691. One of the criteria for settling funding is the number of publications within a given project–and we tried to publish as many of them as possible. The publication cycle within this project is still ongoing, as it turned out that we obtained a surprising number of results (especially in the homogeneous subgroups).

Thank you that the reviewer points this out. We hope that our explanation of listing the bioethics committee number in this way is sufficient.

Reviewer 2 Report

The authors improved the paper, integrating some of my comments, which I appreciated. The sample now is very clear due to the table and also the text improved a lot.

However, some of my comments were not correctly implemented; similarly, now some new issues arose after the table was shown. I do believe there is still some work to do to improve the manuscript.

In brief:

1- Section 2.1 reports that the samples are “matched for age” but this is not true based on the table, which in turn reports a statistical difference in age between controls and patients. The text should be changed. But this suggests also that the analyses should be revised, at least including the age as a covariable/confounder factor, given this difference.

2- Similarly, the newly written exclusion criteria say that patients with comorbid “medical history of psychosis (schizophrenic, affective), significant mood and/or anxiety disorders that required pharmacological treatment” were excluded. However, the table reports that a substantial number of patients also suffer or suffered from MDD, anxiety disorders etc. This should be explained, at least: were such comorbidities not treated (as in the exclusion criteria)? Were they previous diagnoses that are not present at the time of assessment?

Once again, the presence of comorbidities should be carefully taken into account in the analyses, since this might create substantial biases, given that they might indicate different subgroups of patients (e.g., MMD patients might use stimulants as “self-therapy”; on the other hand, anxiety might be caused or elicited by stimulants etc)

Therefore, as I suggested in the previous review, the analyses should be better defined (also in the text) and the possibility of bias should be excluded, by integrating confounders in the analyses. Or at least reporting all these issues as limitations!

3- As in my previous review, the authors did not specify clearly why other studies “suffer from a lack of adequate statistical power”; this should be definitively explained since the discussion with previous literature is very important.

Author Response

Thank you for your comments and remarks. They allowed us to improve the manuscript and, at the same time, take a broader view of the problem under study. Below, in the responses to the following items, we describe where the changes are included.

1 - Section 2.1 reports that the samples are “matched for age” but this is not true based on the table, which in turn reports a statistical difference in age between controls and patients. The text should be changed. But this suggests also that the analyses should be revised, at least including the age as a covariable/confounder factor, given this difference.

- Thank you for this suggestion. Indeed, the reviewer is correct with regard to this point, which we completely agree with. Therefore, we have included an additional table with calculations in the Manuscript. Line261-267, Page7.

2 - Similarly, the newly written exclusion criteria say that patients with comorbid “medical history of psychosis (schizophrenic, affective), significant mood and/or anxiety disorders that required pharmacological treatment” were excluded. However, the table reports that a substantial number of patients also suffer or suffered from MDD, anxiety disorders etc. This should be explained, at least: were such comorbidities not treated (as in the exclusion criteria)? Were they previous diagnoses that are not present at the time of assessment?

Once again, the presence of comorbidities should be carefully taken into account in the analyses, since this might create substantial biases, given that they might indicate different subgroups of patients (e.g., MMD patients might use stimulants as “self-therapy”; on the other hand, anxiety might be caused or elicited by stimulants etc)

Therefore, as I suggested in the previous review, the analyses should be better defined (also in the text) and the possibility of bias should be excluded, by integrating confounders in the analyses. Or at least reporting all these issues as limitations!

- Thank you for this comment. In fact, they were earlier diagnoses resulting from the analysis of the patient's overall records. We highlighted this medical history because of our research integrity.

When recruiting our previous research groups, we also encountered this problem. It's hard to gather a "completely clean" clinical group. We even wondered, while discussing this issue, if such a group existed. The reviewer is right that this factor must be emphasized in the limitations. Therefore, we have added such a note in the text in the limitations section of the study (line 217-419, page.15).

3 - As in my previous review, the authors did not specify clearly why other studies “suffer from a lack of adequate statistical power”; this should be definitively explained since the discussion with previous literature is very important.

- Thank you for this suggestion. We specified the limitations of our own research and those of others in the last paragraph of the discussion (line 419-423, page 15).

Round 3

Reviewer 1 Report

N.A.

Reviewer 2 Report

I thank the authors for improving the comments of my review, which I believe also improved a lot the overall quality of the manuscript.